# Fabrication of Hard Tissue Constructs from Induced Pluripotent Stem Cells for Exploring Mechanisms of Hereditary Tooth/Skeletal Dysplasia

**DOI:** 10.3390/ijms26020804

**Published:** 2025-01-18

**Authors:** Takeru Kondo, Sermporn Thaweesapphithak, Sara Ambo, Koki Otake, Yumi Ohori-Morita, Satomi Mori, Naruephorn Vinaikosol, Thantrira Porntaveetus, Hiroshi Egusa

**Affiliations:** 1Division of Molecular & Regenerative Prosthodontics, Tohoku University Graduate School of Dentistry, Sendai 980-8575, Japan; sara.ambo.d3@tohoku.ac.jp (S.A.); koki.otake.a2@tohoku.ac.jp (K.O.); yumi.ohori.d4@tohoku.ac.jp (Y.O.-M.); satomi.mori.b8@tohoku.ac.jp (S.M.); naruephorn.vinaikosol.p8@dc.tohoku.ac.jp (N.V.); 2Department of Next-Generation Dental Material Engineering, Tohoku University Graduate School of Dentistry, Sendai 980-8575, Japan; 3Center of Excellence in Genomics and Precision Dentistry, Department of Physiology, Clinical Research Center, Faculty of Dentistry, Chulalongkorn University, Bangkok 10330, Thailand; sermporn.t@chula.ac.th (S.T.); thantrira.p@chula.ac.th (T.P.); 4Center for Advanced Stem Cell and Regenerative Research, Tohoku University Graduate School of Dentistry, Sendai 980-8575, Japan

**Keywords:** induced pluripotent stem cell, three-dimensional construct, tooth dysplasia, skeletal dysplasia, dentinogenesis, osteogenesis, hypophosphatasia, pathological phenotype

## Abstract

Tooth/skeletal dysplasia, such as hypophosphatasia (HPP), has been extensively studied. However, there are few definitive treatments for these diseases owing to the lack of an in vitro disease model. Cells differentiated from patient-derived induced pluripotent stem cells (iPSCs) demonstrate a pathological phenotype. This study aimed to establish a method for fabricating hard tissue-forming cells derived from human iPSCs (hiPSCs) for the pathological analysis of tooth/skeletal dysplasia. Healthy (HLTH) adult-derived hiPSCs were cultured in a hard tissue induction medium (HM) with or without retinoic acid (RA) under 3D culture conditions, and mineralization and expression of dentinogenesis- and osteogenesis-related markers in 3D hiPSC constructs were evaluated. hiPSCs derived from patients with hypophosphatasia were also cultured in HM with RA. HLTH-derived hiPSCs formed mineralized 3D constructs and showed increased expression of dentinogenesis- and osteogenesis-related markers; addition of RA promoted the expression of these markers in hiPSC constructs. HPP-derived hiPSC constructs showed lower mineralization and expression of dentinogenesis- and osteogenesis-related markers than HLTH-derived hiPSCs, indicating an impaired ability to differentiate into odontoblasts and osteoblasts. This method for fabricating 3D hiPSC constructs allows for simultaneous assessment of dentinogenesis and osteogenesis, with HPP-derived hiPSC constructs recapitulating pathological phenotypes.

## 1. Introduction

Skeletal dysplasia, such as hypophosphatasia and osteogenesis imperfecta, is a genetic disorder of connective tissues characterized by skeletal deformity and bone fragility [1,2]. Most patients with hereditary skeletal dysplasia present with dental abnormalities such as disturbed dentin formation [3]. Many genes responsible for hereditary tooth and skeletal dysplasias have been identified in recent decades [4]; however, few definitive treatments are available for these hereditary diseases. In vitro models that mimic tooth and skeletal development are required to understand the biological mechanisms of biomineralization and discover new drugs for treatment.

Induced pluripotent stem cells (iPSCs) can be generated by expressing the transcription factors *OCT4*, *SOX2*, *KLF4*, and *MYC* in human somatic cells [5]. Reprogrammed iPSCs closely resemble human embryonic stem cells, and cells differentiated from patient-derived iPSCs can demonstrate the pathological phenotype of a patient [6]. In vitro disease models using patient-specific iPSCs could allow for the elucidation of disease mechanisms along with drug screening to develop novel therapeutic approaches.

Protocols for the in vitro osteogenic differentiation of mouse and human iPSCs are well established [7]. In differentiating iPSCs into osteoprogenitors, embryoid body (EB) formation and mesodermal induction are intermediate steps during osteogenic differentiation [8,9,10,11]. With the development of osteogenic induction protocols for iPSCs, inherited skeletal disease models have also been developed using patient-specific iPSCs [12]. However, there are few studies on the odontoblast differentiation of iPSCs. Although one study reported an odontoblast differentiation method for mouse iPSCs [13], whether this protocol can be applied to human iPSCs (hiPSCs) remains unknown, and disease models using hiPSCs for dentinogenesis imperfecta have not yet been established.

Unlike two-dimensional (2D) cell culture conditions, three-dimensional (3D) cell culture conditions can provide a more complex tissue architecture that is similar to a natural local environment and mimic developmental organogenesis [14]. More accurate 3D organ-like structures are a valuable tool for in vitro disease models [15]. Most previous studies demonstrated the protocols of odontogenic or osteogenic differentiation using iPSCs in 2D culture conditions [8,9,10,11,12,13]. We previously reported that shaking culture can be used to fabricate osteogenically and chondrogenically induced 3D iPSC constructs [16,17]; thus, we used the culture system to achieve odontogenic or osteogenic differentiation of iPSCs in 3D culture conditions.

The media consisting of dexamethasone, β-glycerol phosphate, and ascorbic acid have been widely used to promote odontoblast and osteoblast differentiation of human stem cells [18,19], including hiPSCs [20,21]. Retinoic acid (RA) reportedly promotes the osteoblast differentiation of hiPSCs by shortening the induction period [22]. It has also demonstrated positive effects for odontoblast differentiation [23,24] of stem cells in vitro. Therefore, we hypothesized that the addition of RA to a medium with dexamethasone, β-glycerol phosphate, and ascorbic acid would enhance the dentinogenesis and osteogenesis of 3D hiPSC constructs.

Various markers such as collagen type 1 alpha 1 (COL1A1), dentin matrix acidic phosphoprotein 1 (DMP1), and dentin sialophosphoprotein (DSPP) were used to evaluate odontoblast and osteoblast differentiation [20,21]. These markers could be detected both in odontoblasts and osteoblasts [25]. Since tooth dysplasia often occurs concomitantly with hereditary skeletal dysplasia, simultaneous evaluation of dentinogenesis and osteogenesis as hard tissue-forming cell differentiation by evaluating the above dentinogenesis- and osteogenesis-related markers is valuable for the pathological analysis of hereditary tooth and skeletal dysplasias. Although a previous study evaluated osteogenesis of patient-specific hiPSCs [12], no study has evaluated both dentinogenesis and osteogenesis of the cells. This is the first study to establish a method for fabricating 3D hard tissue constructs derived from hiPSCs that could contribute to the pathological analysis of hereditary tooth and skeletal dysplasias, allowing simultaneous assessment of dentinogenesis and osteogenesis.

## 2. Results

### 2.1. Effects of RA on Mineralization of 3D hiPSC Constructs

To efficiently induce hard tissue-forming cell differentiation of hiPSCs, cells were first cultured in mesodermal induction medium under static conditions (Figure 1a). After 5 d of induction, the cells showed significantly higher expression of T-box transcription factor T (*TBXT*), a mesoderm marker (Figure 1b). Following mesodermal induction, the cells were cultured in a hard tissue induction medium (HM) with or without RA under shaking conditions to form a 3D construct (Figure 1a). Histologically, 3D constructs cultured in HM without RA showed no mineralization after 10 days but showed slight mineralization after 20 days. In contrast, the 3D constructs cultured in HM with RA demonstrated mineralization after 10 days and showed robust mineralization after 20 days (Figure 1c), demonstrating that RA accelerated the hard tissue-forming cell differentiation of hiPSC-derived 3D constructs.

### 2.2. Effects of Retinoic Acid on Dentinogenesis and Osteogenesis of 3D hiPSC Constructs

To investigate the dentinogenesis and osteogenesis of the 3D hiPSC constructs, the expression of four categorized dentinogenesis- and osteogenesis-related genes was quantitatively evaluated. Expression of phosphate metabolism-related genes *DMP1* and phosphate regulating endopeptidase homolog X-linked (*PHEX*), along with extracellular matrix (ECM)-related genes *DSPP* and *COL1A1*, in 3D hiPSC constructs gradually increased during hard tissue induction without RA. The addition of RA further promoted the expression of these genes in the 3D constructs during hard tissue induction. The expressions of *PHEX* and *DSPP* in the RA addition groups were 2–3 times higher than those in the HM-only groups. The expressions of *DMP1* and *COL1A1* in the RA addition groups were approximately five and eight times higher than those in the HM-only groups, respectively (Figure 2a,b). Similarly, HM on its own slightly promoted the expression of WNT signaling genes WNT inhibitory factor 1 (*WIF1*) and sclerostin (*SOST*), as well as transcription factors msh homeobox 1 (*MSX1*), sp7 transcription factor (*SP7*), paired box gene 9 (*PAX9*), and LIM homeobox domain 6 (*LHX6*) in 3D hiPSC constructs, and RA addition significantly upregulated the expression of these genes in the constructs. The expressions of *WIF1*, *SOST*, *PAX9*, and *LHX6* were approximately 2–3 times higher for 3D hiPSC constructs cultured in HM with RA than in those cultured in HM without RA. The expressions of *MSX1* and *SP7* in the RA addition groups were approximately 6 and 3000 times higher than those in the HM-only groups, respectively (Figure 2c,d).

The expression of dentinogenesis- and osteogenesis-related proteins was evaluated by immunohistochemical staining. The 3D hiPSC constructs cultured in HM with RA demonstrated strong expression of DMP1, DSPP, WIF1, and MSX1 compared to those cultured in HM without RA (Figure 3). These results indicate that HM with RA promoted odontoblast and osteoblast differentiation of the hiPSC-derived 3D constructs.

### 2.3. Mineralization of 3D Constructs Fabricated from Hypophosphatasia (HPP) Patient-Derived hiPSCs

To apply the established dentinogenesis and osteogenesis induction methods using iPSCs to exhibit dentinogenesis and osteogenesis imperfecta, hiPSCs derived from patients with hypophosphatasia (HPP) were generated. The morphologies of both HPP- and healthy (HLTH) adult-derived hiPSCs were similar under a microscope (Figure 4a), and there was no statistical difference in the expression of *TBXT* between the HPP and HLTH groups after mesodermal induction (Figure 4b). In contrast, the 3D constructs in the HPP group showed impaired mineralization compared to those in the HLTH group after hard tissue induction with RA (Figure 4c). These results suggest that HPP patient-derived hiPSCs cannot differentiate into hard tissue-forming cells as HLTH-derived hiPSCs can.

### 2.4. Dentinogenesis and Osteogenesis of 3D Constructs Fabricated from HPP-Derived hiPSCs

To investigate dentinogenesis and osteogenesis in HPP-derived hiPSCs, the expression of dentinogenesis- and osteogenesis-related genes was evaluated after hard-tissue induction with RA. The 3D constructs in the HPP group showed significantly lower expression of phosphate metabolism-related genes (*DMP1* and *PHEX*), ECM-related genes (*DSPP* and *COL1A1*), regulators of WNT signaling (*MSX1* and *SP7*), and transcription factors (*PAX9* and *LHX6*) than those in the HLTH group (Figure 5a–d). Histological analysis demonstrated that the protein expression levels of DMP1, DSPP, WIF1, and MSX1 in the 3D hiPSC constructs were lower in the HPP group than in the HLTH group (Figure 5). Thus, 3D constructs formed by HPP-derived hiPSCs demonstrated remarkably impaired mineralization and lower expression of dentinogenesis- and osteogenesis-related molecules compared with those formed by HLTH-derived hiPSCs. These results indicate that HPP-derived hiPSCs would lack the ability to differentiate into odontoblasts and osteoblasts during developmental organogenesis of hard tissue, which recapitulates the disease phenotypes of patients with HPP.

## 3. Discussion

We established a method to assess the differentiation of hiPSCs into odontoblasts and osteoblasts using 3D constructs. Notably, 3D hiPSC constructs derived from patients with HPP demonstrated impaired odontoblast and osteoblast differentiation compared with those derived from healthy adults (Figure 5 and Figure 6), indicating the recapitulation of the disease phenotypes of patients with HPP in our system.

Specific markers of odontoblasts have not yet been identified because the molecules involved in dentinogenesis are also involved in osteogenesis; however, the crucial molecules and biological mechanisms underlying dentin formation have been identified. In this study, we categorized four types of dentinogenesis- and osteogenesis-related genes and proteins. DSPP is a member of the small integrin-binding ligand N-linked glycoprotein (SIBLING) family and is primarily expressed in odontoblasts. The DSPP signal transcript encodes dentin phosphoprotein and dentin sialoprotein, which are the major dentin-specific proteins [26]. Dspp^−/−^ mice demonstrate tooth dysplasia such as sporadic unmineralized areas within the dentin and frequent pulp exposure, similar to those observed in patients with dentinogenesis imperfecta [27]. *Dspp* overexpression in Dspp^−/−^ mice rescues dentinogenesis imperfecta in Dspp^−/−^ mice [28]. DMP1, another member of the SIBLING family, was originally detected in rat dentin [29]. DMP1 is located at the same genomic locus as DSPP and has similar biological features [30]. Deletion of *Dmp1* leads to the partial failure of odontogenesis and mineralization [31]. Dmp1^−/−^ mice have reduced dentin walls, dentin hypomineralization, abnormalities in dentinal tubes, and enlarged pulp chambers [27]. Re-expression of Dmp1 rescued dentinogenesis in Dmp1^−/−^ mice [32]. These findings indicate that DSPP and DMP1 are crucial regulators of odontoblast differentiation and dentin formation. The 3D constructs formed by HPP patient-derived hiPSCs showed significantly lower expression of DSPP and DMP1 compared to those formed by healthy adult-derived hiPSCs (Figure 4 and Figure 5), indicating in vitro recapitulation of dentinogenesis imperfecta in HPP patients.

MSX1, PAX9, and LHX6 are crucial transcription factors involved in tooth germ development. *Msx1*, *Pax9*, and *Lhx6* are expressed in dental mesenchyme during odontogenesis [33,34]. Knockout mutation of *Msx1* leads to impaired tooth development in mice [35], and *Pax9* knockout mice arrested tooth germ development [36]. *Lhx6*-mutant mice show an absence of molars due to failure in the specification of the molar mesenchyme [37]. Wnt signaling regulates tooth germ development; Wif1, a Wnt antagonist, was detected in the dental mesenchyme during early tooth germ development [38]. Overexpression of *Wif1* promotes the dentinogenesis of stem cells from the apical papilla [39], indicating that WIF1 is a regulator of tooth development. These molecules can be used as markers of tooth development. In this study, the expression of these molecules increased in 3D hiPSC constructs during hard tissue induction (Figure 2 and Figure 3). These results suggest that the 3D hiPSC constructs contain cells related to tooth germ development, as well as osteogenic progenitors, although MSX1 [35], PAX9 [40], LHX6 [34], and WIF1 [41] are also important regulators of craniofacial skeletal development.

RA is a critical molecule in chordate embryonic development [42]. RA signaling contributes to the development of neuromesodermal precursors in the embryo, postnatal bone, and teeth [43]. Interruption of RA signaling induces defects in various organs, including forelimbs and teeth [44,45]. RA receptors are expressed in stem and osteoblast-lineage cells, and RA signaling plays an important role in the cellular differentiation of osteoblasts and odontoblasts [43,46]. This study also demonstrated that RA promoted osteoblast and odontoblast differentiation in mesoderm-induced hiPSCs (Figure 2 and Figure 3). RA reportedly regulates BMP and Wnt signaling [47,48]. Both BMP and Wnt signaling are crucial for skeletal and tooth development [49,50] and can promote osteoblast and odontoblastic differentiation [51,52]. RA has also been reported to enhance osteogenic differentiation by activating BMP and Wnt signaling [22]. RA may promote osteoblast and odontoblast differentiation of cells within the 3D hiPSC constructs by regulating BMP and Wnt signaling; however, further in vitro analysis is needed to confirm this.

HPP is a rare hereditary disease characterized by impaired bone and tooth development [2]. It is caused by mutations in the *ALPL* gene that encodes tissue-nonspecific alkaline phosphatase (ALP) [53]. ALP hydrolyzes inorganic pyrophosphate to generate inorganic phosphate, which is critical for cellular mineralization [54]. Impaired ALP activity in patients with HPP suppresses osteogenic differentiation of mesenchymal stem cells [55]. In addition, impaired odontoblast differentiation due to ALP mutations results in disturbed dentinogenesis [56]. Enzyme replacement therapy using recombinant ALP restores ALP activity and improves the clinical symptoms in patients with HPP [57]. However, this therapy has several drawbacks, such as frequent subcutaneous injections, allergic reactions, and high costs [58]. Further exploration of the disease mechanism behind HPP and the establishment of alternative therapies are required. Our established system using HPP-derived hiPSCs may contribute to the investigation of the molecular mechanisms underlying HPP and the development of novel therapies.

A crucial advantage of in vitro models is that they can be used for drug screening, such as high-throughput screening. Drug screening using a large number of chemicals requires an assessment system with a shorter time, lower cost, and convenience [59]. In our system, dentinogenesis and osteogenesis of hiPSCs were simultaneously evaluated in a 3D construct within 14 days of hard tissue induction (Figure 1, Figure 2 and Figure 3). In addition, only minimal chemicals such as RA were necessary for cell culture. To expand this model to a high-throughput screening platform, a technology to visualize gene expression in hiPSCs using a reporter system should be established in the future. High-throughput drug screening using this model with hiPSCs derived from patients with HPP and those with other hereditary tooth and skeletal dysplasias will contribute to the development of novel therapeutic technologies for these diseases.

The in vitro assessment model developed in this study also has a significant limitation. In our model, it was impossible to distinguish between dentinogenesis and osteogenesis within a 3D construct because the evaluated markers and the components of the HM were involved in both differentiation processes. Further improvements to the assessment of differentiation at the cellular level are required. Another key limitation of this study is the use of a single HPP-derived hiPSC line. Future studies should validate these findings using multiple hiPSC lines derived from other patients with HPP to ensure generalizability and confirm the reproducibility of our protocol.

In conclusion, this study proposed a system for assessing dentinogenesis and osteogenesis using hiPSC-derived 3D constructs. This system, using hiPSCs derived from patients with hereditary skeletal or tooth dysplasia, may shed light on the unrecognized mechanisms of these diseases and contribute to the discovery of new drugs for their treatment.

## 4. Materials and Methods

### 4.1. hiPSC Culture

hiPSCs derived from a healthy adult (HPS0076) and from a patient with HPP (HPS0332) were provided by the RIKEN BRC through the National Bio-Resource Project of the Mext, Japan. HPP-derived hiPSCs were generated using four transcription factors, Oct3/4, Sox2, Klf4, and cMyc, by a female infant with the variant in the ALPL gene. hiPSCs were maintained on mitomycin C-treated SNLP76.7-4 feeder cells in embryonic stem cell medium composed of Primate ES Cell Medium (Reprocell, Tokyo, Japan) supplemented with 4 ng/mL basic fibroblast growth factor (Reprocell). The medium was changed every other day.

The hiPSCs were passaged onto iMatrix-511 silk (Nippi, Tokyo, Japan)-coated feeder-free plates. The cells were cultivated in an embryonic stem medium composed of StemFit^®^AK02N (Reprocell) supplemented with 10 µM CultureSure^®^ Y-27632 (Fuji Film, Tokyo, Japan). The medium without Y-27632 was changed on the day after passaging and every other day thereafter. hiPSCs were passaged every 7 days using an equal volume of TrypLETM Select (Gibco, Waltham, MA, USA) and 0.5 mmol/L EDTA/PBS Solution (Nacalai Tesque, Kyoto, Japan).

### 4.2. Mesodermal Induction and Hard Tissue Induction of hiPSCs

hiPSCs were transferred into a 24-well Elplasia^®^ plate (Corning, Corning, NY, USA) in the embryonic stem cell medium and allowed to form EBs overnight. Afterward, the entire medium was replaced with mesodermal induction medium [33.75% Dulbecco’s Modified Eagle Medium (DMEM)/F12, 48.25% Neurobasal medium supplemented with 1X N2 supplement (Gibco), 1X B-27 supplement (Gibco), 0.1% BSA, 5 µM cyclopamine (Gibco), and 30 µM CHIR (Funakoshi, Tokyo, Japan)].

On Day 5 of mesodermal differentiation, the EBs were transferred from four wells of the 24-well Elplasia^®^ plate to a 25T flask and were cultured in hard tissue induction medium [DMEM high glucose (Nacalai Tesque), supplemented with 15% fetal bovine serum (FBS; Thermo Fisher Scientific, Waltham, MA, USA), 10 nM dexamethasone (Sigma-Aldrich, Waltham, MA, USA), 10 mM β-glycerophosphate disodium salt hydrate (Sigma-Aldrich), 50 ng/mL L-ascorbic acid 2-phosphate (Sigma-Aldrich), and 250 ng/mL antibiotic-antimycotic (Gibco)] with or without 0.1 µM RA (Sigma-Aldrich) on a seesaw shaker (BIO CRAFT, Tokyo, Japan). Half of the medium was replaced every other day.

### 4.3. Histological Staining

#### 4.3.1. Hematoxylin and Eosin Staining

The collected 3D constructs were fixed using 4% paraformaldehyde (pH 7.4) and embedded in paraffin. The blocks were cut into 5 μm thick sections and mounted on glass slides. For H&E staining, slides were deparaffinized with xylene and rehydrated using a graded alcohol series (100%, 95%, and 70%). The slides were then washed with distilled water and stained with hematoxylin solution (Muto Pure Chemicals, Tokyo, Japan) for 5 min. The slides were then washed under running tap water for 5 min and counterstained with eosin Y solution (Muto Pure Chemicals) for 1 min. Stained slides were dehydrated in graded alcohol (70%, 95%, and 100%) and cleared in xylene twice for 5 min each. The slides were mounted in malinol (Muto Pure Chemicals). Images were captured under a light microscope (Nikon, Tokyo, Japan).

#### 4.3.2. Von Kossa Staining

The slides were deparaffinized and then stained by incubating them with 1.8 mM fast red TR (Sigma) and 0.9 mM naphthol AS-MX phosphate (Sigma-Aldrich) in 120 mM Tris buffer (pH 8.4) for 30 min at 37 °C. Nodule mineralization was visualized by treatment with 5% silver nitrate solution (Sigma-Aldrich) under ultraviolet light for 30 min, followed by 5% sodium thiosulfate solution (Sigma-Aldrich) for 5 min. Images were captured using a light microscope (Nikon).

#### 4.3.3. Immunohistochemistry

For immunostaining, slides were deparaffinized and heated in an autoclave at 80 °C for 30 min. Antigen retrieval was performed using Tris-ethylenediaminetetraacetic acid (pH 9.0). The sections were then treated with 3% H_2_O_2_ to eliminate endogenous peroxidase activity. Next, the slides were incubated overnight at 4 °C with primary antibodies against DMP1 (Santa Cruz Biotechonology, Dallas, TX, USA), DSPP (Santa Cruz Biotechonology), WIF1 (Santa Cruz Biotechonology), and MSX1 (Santa Cruz Biotechonology), followed by incubation for 1 h with Pierce^®^ Goat Anti-Rat IgG (H + L), Peroxidase Conjugated (Thermo Fisher Scientific) as the secondary antibody. The signal was detected using 3,3′-diaminobenzidine tetrahydrochloride (Roche Applied Science, Mannheim, Germany), followed by counterstaining with Mayer’s hematoxylin solution (Wako, Osaka, Japan) for 5 s. Images were captured using a light microscope.

### 4.4. Reverse Transcription-Polymerase Chain Reaction

Total RNA was extracted from EBs using the RNeasy Mini Kit (QIAGEN, Germantown, MD, USA) and quantified using a Thermo Scientific NanoDrop 1000 ultraviolet-visible spectrophotometer (NanoDrop Technologies, Wilmington, DE, USA). After treatment with DNase I (Thermo Fisher Scientific), cDNA was synthesized using the PrimerScript 1st Strand cDNA Synthesis Kit (Takara Bio, Shiga, Japan).

mRNA expression was measured using the StepOnePlus Real-Time PCR system (Thermo Fisher Scientific) and the Thunderbird SYBR qPCR Mix (Toyobo, Osaka, Japan) for SYBR green-based PCR, or TaqManTM Gene Expression Master Mix (Thermo Fisher Scientific) for TaqMan probe-based PCR. Glyceraldehyde 3-phosphate dehydrogenase *GAPDH* was used as a reference gene. Each experiment was performed in triplicate, and relative mRNA expression was calculated using the 2^−ΔΔCq^ method.

### 4.5. Statistics and Reproducibility

The Student’s *t*-test and one-way analysis of variance with Tukey’s multiple comparison test were used for statistical analyses. Statistical significance was set at *p* < 0.05. The sample sizes and numbers of replicates are reported in the figure legends. Data are presented as means and standard deviations.

## Figures and Tables

**Figure 1 ijms-26-00804-f001:**
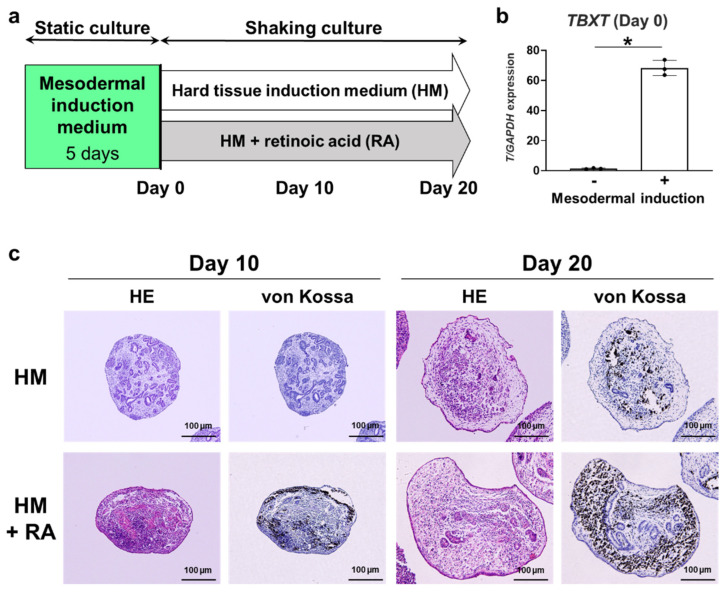
Effects of retinoic acid on mineralization of 3D human induced pluripotent stem cell (hiPSC) constructs during hard tissue induction. (**a**) Cell culture and 3D hard tissue induction methods for hiPSCs. “ES medium” is growth medium for hiPSCs. “Day 0” refers to the day when hard tissue induction commences. (**b**) Expression of mesoderm marker *TBXT* at Day 0 was determined by quantitative real-time RT-PCR analysis (*n* = 3). (**c**) Representative hematoxylin and eosin (HE) and von Kossa staining images of 3D hiPSC constructs, cultured in HM with or without RA under shaking conditions, on Day 10 and Day 20 (scale bars: 100 μm). Student’s *t*-test. Data are presented as mean ± SD; * *p* < 0.05 was considered significant.

**Figure 2 ijms-26-00804-f002:**
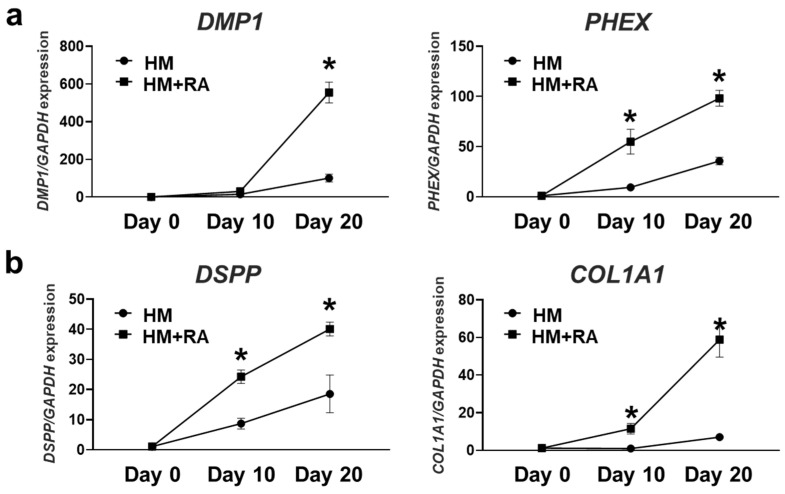
Effects of retinoic acid on expression of dentinogenesis- and osteogenesis-related genes of 3D hiPSC constructs during hard tissue induction. (**a**) Expression of phosphate metabolism-related genes such as dentin matrix acidic phosphoprotein 1 (*DMP1*) and phosphate regulating endopeptidase homolog X-linked (*PHEX*) was determined by quantitative real-time RT-PCR analysis (*n* = 3). (**b**) Expression of ECM-related genes such as dentin sialophosphoprotein (*DSPP*) and *collagen type 1 alpha 1* (*COL1A1*) was determined by quantitative real-time RT-PCR analysis (*n* = 3). (**c**) Gene expression of regulators of WNT signaling, such as WNT inhibitory factor 1 (*WIF1*) and sclerostin (*SOST*), was determined by quantitative real-time RT-PCR analysis (*n* = 3). (**d**) Gene expression of transcription factors such as *msh homeobox 1* (*MSX1*), sp7 transcription factor (*SP7*), paired box gene 9 (*PAX9*), and LIM homeobox domain 6 (*LHX6*) was determined by quantitative real-time RT-PCR analysis (*n* = 3). Student’s *t*-test at the same time point. Data are presented as mean ± SD; * *p* < 0.05 was considered significant.

**Figure 3 ijms-26-00804-f003:**
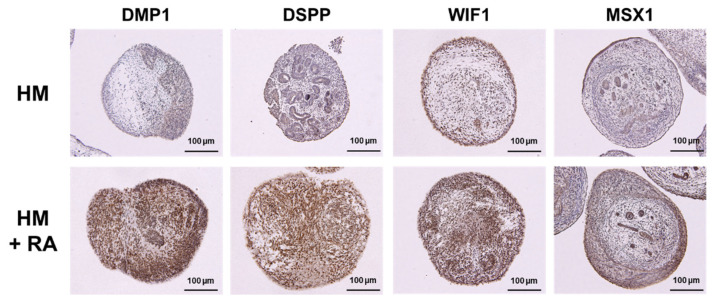
Effects of retinoic acid on dentinogenesis- and osteogenesis-related protein expression of 3D hiPSC constructs after hard tissue induction. Three-dimensional hiPSC constructs were assessed on Day 20 of hard tissue induction with or without RA by immunohistochemical staining for DMP1, DSPP, WIF1, and MSX1 (scale bars: 100 μm).

**Figure 4 ijms-26-00804-f004:**
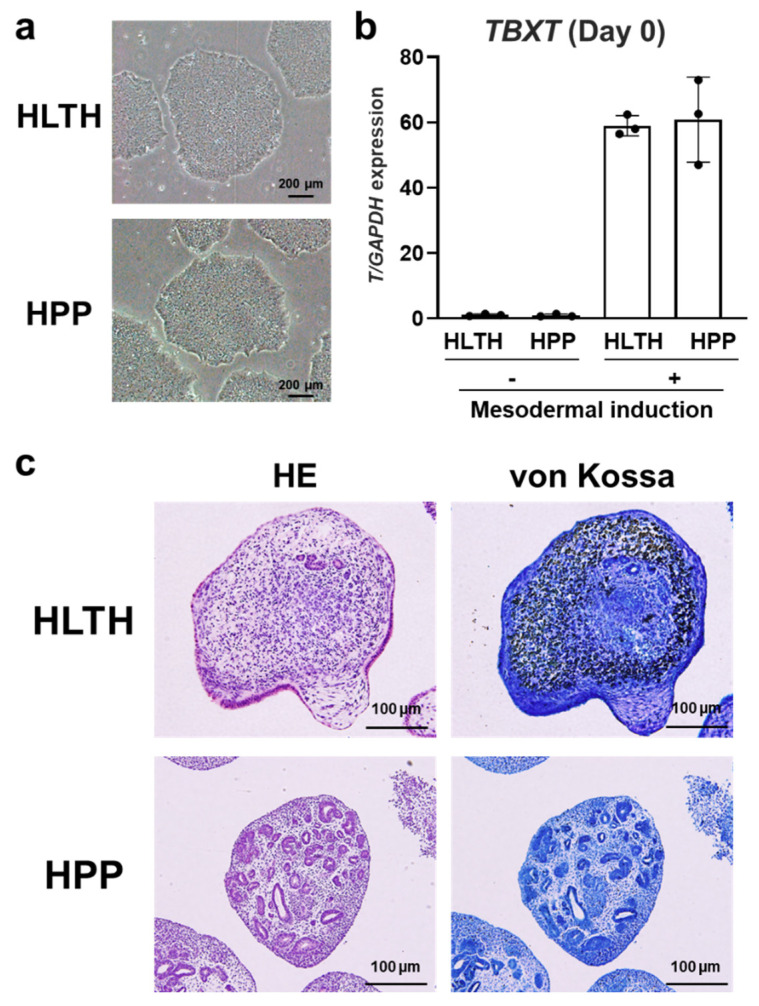
Mineralization of 3D constructs fabricated from hiPSCs from a patient with hypophosphatasia (HPP) after hard tissue induction. (**a**) Representative bright-field microscopic images of healthy (HLTH) adult- and HPP-derived hiPSCs (scale bars: 200 μm). (**b**) Expression of *TBXT* at Day 0 was determined by quantitative real-time RT-PCR analysis (*n* = 3). (**c**) Representative HE and von Kossa staining images of 3D constructs fabricated from HLTH and HPP-derived hiPSCs, cultured in HM with RA for 20 days (scale bars: 100 μm). Student’s *t*-test at the same time point. There were no significant differences between the groups at the same time points. Data are presented as mean ± SD; *p* < 0.05 was considered significant.

**Figure 5 ijms-26-00804-f005:**
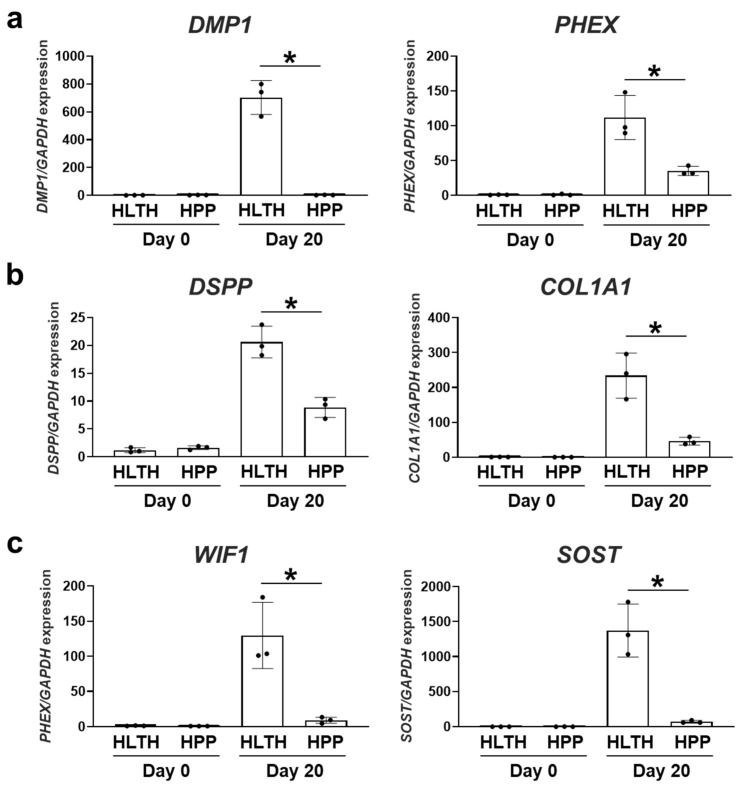
Expression of dentinogenesis- and osteogenesis-related genes of 3D constructs fabricated from HPP patient-derived hiPSCs after hard tissue induction with RA. (**a**) Gene expression of *DMP1* and *PHEX* was determined by quantitative real-time RT-PCR analysis (*n* = 3). (**b**) Gene expression of *DSPP* and *COL1A1* was determined by quantitative real-time RT-PCR analysis (*n* = 3). (**c**) Gene expression of *WIF1* and *SOST* was determined by quantitative real-time RT-PCR analysis (*n* = 3). (**d**) Gene expression of *MSX1*, *SP7*, *PAX9*, and *LHX6* was determined by quantitative real-time RT-PCR analysis (*n* = 3). Student’s *t*-test at the same time point. Data are presented as mean ± SD; * *p* < 0.05 was considered significant.

**Figure 6 ijms-26-00804-f006:**
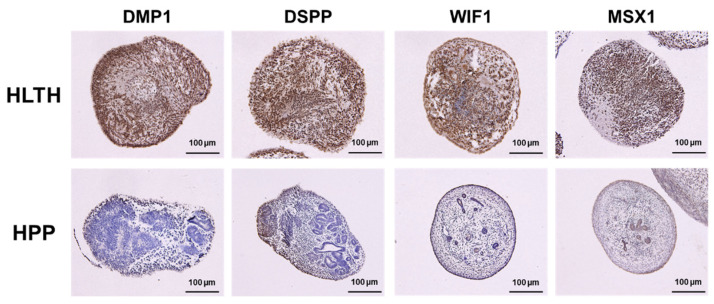
Expression of dentinogenesis- and osteogenesis-related proteins in 3D constructs fabricated from HPP-derived hiPSCs after hard tissue induction with RA. 3D hiPSC constructs were assessed on Day 20 of hard tissue induction with RA by immunohistochemical staining for DMP1, DSPP, WIF1, and MSX1 (scale bars: 100 μm).

## Data Availability

The original contributions presented in this study are included in the article. Further inquiries can be directed to the corresponding authors.

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
