# Peer review of "Fabrication of Hard Tissue Constructs from Induced Pluripotent Stem Cells for Exploring Mechanisms of Hereditary Tooth/Skeletal Dysplasia"

_ijms, 2025, doi:10.3390/ijms26020804_

Round 1

Reviewer 1 Report

Comments and Suggestions for Authors

This work studies a method for fabricating hard 25 tissue-forming cell derived human iPSCs. It is well-written. Here are some comments to enhance the quality of the manuscript: 

- Additional emphasis on the novelty compared to previous studies would strengthen the introduction.

- Some captions lack sufficient detail (e.g., Figure 1c). Include more descriptive explanations of the experiments or conditions depicted.

- In the introduction lines 50-54: Expand on how iPSC models compare to other in vitro systems (e.g., organoids, 2D cultures) in simulating hereditary conditions.

- More comparative metrics (e.g., percentage increases in marker expression) would help readers quantify the effect of RA.

Highlight how findings differ significantly from healthy constructs, as this is crucial for validating the model.

-The potential for drug screening could be expanded to discuss future integration with high-throughput screening platforms.

Comments on the Quality of English Language

Perfect!

Author Response

Additional emphasis on the novelty compared to previous studies would strengthen the introduction.

Response: We appreciate your accurate assessment of our research. Following your suggestion, we have added several sentences and revised the introduction as follows:

“Most previous studies demonstrated the protocols of odontogenic or osteogenic differentiation using iPSCs in 2D culture conditions [8-13]. We previously reported that shaking culture can be used to fabricate osteogenically and chondrogenically induced 3D iPSC constructs [16, 17]; thus, we used the culture system to achieve odontogenic or osteogenic differentiation of iPSCs in 3D culture conditions” (Lines 71-76).

“Although a previous study evaluated osteogenesis of patient-specific hiPSCs [12], no study has evaluated both dentinogenesis and osteogenesis of the cells. This is the first study to establish a method for fabricating 3D hard tissue constructs derived from hiPSCs that could contribute to the pathological analysis of hereditary tooth and skeletal dysplasias, allowing simultaneous assessment of dentinogenesis and osteogenesis” (Line number 91-96).

Some captions lack sufficient detail (e.g., Figure 1c). Include more descriptive explanations of the experiments or conditions depicted.

Response: Thank you for your valuable suggestion. As you pointed out, we have revised the captions of Figures 1c, 3, 4c, and 6.

In the introduction lines 50-54: Expand on how iPSC models compare to other in vitro systems (e.g., organoids, 2D cultures) in simulating hereditary conditions.

Response: We appreciate your thoughtful comments. Following your recommendation, we have added the following sentences to the introduction:

“Unlike two dimensional (2D) cell culture conditions, three-dimensional (3D) cell culture conditions can provide a more complex tissue architecture that is similar to a natural local environment and mimic developmental organogenesis [14]. More accurate 3D organ-like structure is a valuable tool for in vitro disease models [15]” (Lines 68-71).

More comparative metrics (e.g., percentage increases in marker expression) would help readers quantify the effect of RA.

Response: We are grateful for your valuable suggestion. Based on your comments, we have added the sentences describing percentage increases to the results section as follows:

“The expressions of PHEX and DSPP in the RA addition groups were 2-3 times higher than those in the HM only groups. The expressions of DMP1 and COL1A1 in the RA addition groups were approximately five and eight times higher than those in only HM groups, respectively” (Lines 128-132).

“The expressions of WIF1, SOST, PAX9, and LHX6 were approximately 2-3 times higher for 3D hiPSC constructs cultured in HM with RA than in those cultured in HM without RA. The expressions of MSX1 and SP7 in the RA addition groups were approximately 6 and 3000 times higher than those in the HM only groups, respectively” (Lines 136-140).

Highlight how findings differ significantly from healthy constructs, as this is crucial for validating the model.

Response: Thank you for your suggestion. To clear the difference between the 3D constructs formed by HPP- and HLTH-derived hiPSCs, we have added the following sentences to the results:

“Thus, 3D constructs formed by HPP-derived hiPSCs demonstrated remarkably impaired mineralization and lower expression of dentinogenesis- and osteogenesis-related molecules compared with those formed by HLTH-derived hiPSCs. These results indicate that HPP-derived hiPSCs would lack the ability to differentiate into odontoblasts and osteoblasts during developmental organogenesis of hard tissue, which recapitulates the disease phenotypes of patients with HPP” (Line number 193-198).

Moreover, to emphasize the significance of the establishment of this model, we have added the following sentence to the initial paragraph of the discussion:

“Notably, 3D hiPSC constructs derived from patients with HPP demonstrated impaired odontoblast and osteoblast differentiation compared with those derived from healthy adults (Figures 5 and 6), indicating the recapitulation of the disease phenotypes of patients with HPP in our system” (Lines 217-220).

The potential for drug screening could be expanded to discuss future integration with high-throughput screening platforms.

Response: Thank you for your suggestion. We have revised the discussion about drug screening as follows:

“To expand this model to a high-throughput screening platform, a technology to visualize gene expression in hiPSCs using a reporter system should be established in the future. High-throughput drug screening using this model with hiPSCs derived from patients with HPP and those with other hereditary tooth and skeletal dysplasias will contribute to the development of novel therapeutic technologies for these diseases” (Line number 287-292).

Reviewer 2 Report

Comments and Suggestions for Authors

Drs Kondo and colleagues work shows that some trails of the hypophosphatasia (HPP) disease can be replicated in vitro using an hiPSC line carrying mutations that give rise to this disease.  They showed that compared to healthy cell lines, the HPP cell lines do not mineralized 3D constructs equally to control groups.  In addition, they observed significant differences in the expression of genes between groups related to dentin formation, indicating the HPP-derived constructs recapitulate the pathological phenotypes of HPP.  Although, the data support this conclusion, the following issues must be addressed to verify the results and to provide all the information required for readers to replicate these findings.

1- The authors perform the experiments with one single HPP cell line, and do not provide information of the genetic background of the parental cell line.  Thus, it will be required to provide details information and characterization of the HPP cell lines  and also if possible, the same experiments should be performed with a second/third line with the same genetic background. This will validated their findings and will allow others to replicate their results. 

2- The authors tried to evaluate the effect of HPP in dentinogenesis and clarify that this is difficult because similar genetic profile is observed between dentoblasts and osteoblasts.  Thus, it is difficult to convince the reader that the data provided here relates more to the pathogenesis observed in dentogenesis.  One suggestion will be to do side comparative studies of 3D constructs differentiating specifically in osteoblast, and compared to the current 3D constructs developed in the HM medium with and without RA.  The comparative genetic analysis should indicate differences in expression between the groups and thus, the interpretation towards dentogenesis should be clear. 

3. Although not the main goal of this manuscript, it will be useful and required to provide details characterization of the HPP hiPSC lines and compare to control cells lines, to determine potential phenotypical differences between both cell lines in undifferentiated state and in their mesoderm stage before going to odontogenesis or blastogenesis. 

Author Response

Response to Reviewer #2 Comments

1- The authors perform the experiments with one single HPP cell line, and do not provide information of the genetic background of the parental cell line. Thus, it will be required to provide details information and characterization of the HPP cell lines and also if possible, the same experiments should be performed with a second/third line with the same genetic background. This will validated their findings and will allow others to replicate their results. 

Response: We thank you for assessing our new investigation. Following your suggestion, we have added the information about HPP-derived hiPSCs as follows:

“HPP-derived hiPSCs were generated using four transcription factors, Oct3/4, Sox2, Klf4, and cMyc, by a female infant with the variant in ALPL gene” (Lines 312-314).

While we acknowledge the importance of utilizing multiple cell lines to validate our findings and enhance reproducibility, resource constraints within the scope of this study prevented us from incorporating additional lines. We recognize this as a limitation and have explicitly stated the following: "Another key limitation of this study is the use of a single HPP-derived hiPSC line. Future studies should validate these findings using multiple hiPSC lines derived from other patients with HPP to ensure generalizability and confirm the reproducibility of our protocol" (Lines 297-300).

2- The authors tried to evaluate the effect of HPP in dentinogenesis and clarify that this is difficult because similar genetic profile is observed between dentoblasts and osteoblasts. Thus, it is difficult to convince the reader that the data provided here relates more to the pathogenesis observed in dentogenesis. One suggestion will be to do side comparative studies of 3D constructs differentiating specifically in osteoblast and compared to the current 3D constructs developed in the HM medium with and without RA. The comparative genetic analysis should indicate differences in expression between the groups and thus, the interpretation towards dentogenesis should be clear. 

Response: We appreciate your valuable suggestion. If we could perform your suggested experiments, the results must address the differences between dentinogenesis and osteogenesis. However, the components of odontoblast and osteoblast differentiation medium are quite similar [1-7], and it is impossible to distinguish between odontoblast and osteoblast differentiation medium. Therefore, we have mentioned the importance of the markers evaluated in this study, such as DSPP and DMP1 in odontoblast differentiation, in the discussion section. Moreover, we described this concern in the discussion section as follows:

“In our model, it was impossible to distinguish between dentinogenesis and osteogenesis within a 3D construct, because the evaluated markers and the components of the HM were involved in both differentiation processes. Further improvements to the assessment of differentiation at the cellular level are required” (Lines 294-297).

[1]          Ko JY, Park S, Im GI. Osteogenesis from human induced pluripotent stem cells: an in vitro and in vivo comparison with mesenchymal stem cells. Stem Cells and Development 2014;23:1788–1797.

[2]          Egusa H, Kayashima H, Miura J, Uraguchi S, Wang F, Okawa H, et al. Comparative analysis of mouse-induced pluripotent stem cells and mesenchymal stem cells during osteogenic differentiation in vitro. Stem Cells and Development 2014;23:2156–2169.

[3]          Pan H, Yang Y, Xu H, Jin A, Huang X, Gao X, et al. The odontoblastic differentiation of dental mesenchymal stem cells: molecular regulation mechanism and related genetic syndromes. Frontiers in Cell and Developmental Biology 2023;11:1174579.

[4]          Yang Y, Zhao Y, Liu X, Chen Y, Liu P, Zhao L. Effect of SOX2 on odontoblast differentiation of dental pulp stem cells. Molecular Medicine Reports 2017;16:9659–9663.

[5]          Yin X, Zeng Z, Xing J, Zhang A, Jiang W, Wang W, et al. Hey1 functions as a positive regulator of odontogenic differentiation in odontoblast‑lineage cells. International Journal of Molecular Medicine 2018;41:331–339.

[6]          Khazaei S, Khademi A, Nasr Esfahani MH, Khazaei M, Nekoofar MH, Dummer PMH. Isolation and differentiation of adipose-derived stem cells into odontoblast-like cells: A preliminary in vitro study. Cell Journal 2021;23:288–293.

[7]          Wu Q, Li J, Song P, Chen J, Xu Y, Qi S, et al. Knockdown of NRAGE induces odontogenic differentiation by activating NF-κB signaling in mouse odontoblast-like cells. Connective Tissue Research 2019;60:71–84.

  1. Although not the main goal of this manuscript, it will be useful and required to provide details characterization of the HPP hiPSC lines and compare to control cells lines, to determine potential phenotypical differences between both cell lines in undifferentiated state and in their mesoderm stage before going to odontogenesis or blastogenesis. 

Response: Thank you for your comment. The cell quality and pluripotency of the HPP-derived hiPSCs were validated by the providers (RIKEN BRC). The cell morphologies of both HPP and HLTH-derived hiPSCs were similar as shown in Figure 4a, and there was no difference in cell proliferation between HPP and HLTH-derived hiPSCs. Furthermore, there was no statistical difference in the expression of mesodermal marker TBXT in the mesodermal stage between the HPP and HLTH groups before hard tissue induction as shown in Figure 4b. Thus, there was no difference in the undifferentiated state and the differentiation potential other than that of odontoblast and osteoblast differentiation.

Round 2

Reviewer 2 Report

Comments and Suggestions for Authors

No comments